# Identifying the Barriers to Universal Cervical Length Screening for Preterm Birth Prevention at a Tertiary Hospital in Thailand (Physician Perspectives): Implementation Research

**DOI:** 10.3390/healthcare11071039

**Published:** 2023-04-04

**Authors:** Saifon Chawanpaiboon, Vitaya Titapant, Sanitra Anuwutnavin, Attapol Kanjanapongporn, Julaporn Pooliam

**Affiliations:** 1Division of Maternal-Fetal Medicine, Department of Obstetrics & Gynecology, Faculty of Medicine, Siriraj Hospital, Mahidol University, Bangkok 10700, Thailand; vitaya.tit@mahidol.ac.th (V.T.); asanitra@hotmail.com (S.A.); 2Department of Social Sciences, Faculty of Social Sciences and Humanities, Mahidol University, Nakhon Pathom 73170, Thailand; attapol.kan@mahidol.ac.th; 3Clinical Epidemiological Unit, Office for Research and Development, Faculty of Medicine, Siriraj Hospital, Mahidol University, Bangkok 10700, Thailand; juscma@hotmail.com

**Keywords:** barriers, physician perspective, preterm birth, prevention, universal cervical length screening

## Abstract

Objective: To identify physicians’ views on the barriers to measuring cervical length for preventing preterm deliveries. Materials and methods: This prospective, descriptive implementation study had three phases. In Phase I, 20 physicians were interviewed. Phase II comprised questionnaire development and data validation. The questionnaire was distributed to 120 Phase III participants. Results and discussion: All 120 participants responded. In 44 cases, the physicians received support from their local Maternal and Child Health Boards for preterm-birth-prevention programs; the other 76 physicians did not. The doctors tended to believe that cervical length screening plays no role in preventing preterm births (4/44 (9.1%) and 24/76 (31.6%); OR, 4.615; 95% CI, 1.482–14.373; *p* = 0.005). They were unsure about the correct measurement procedures (13/44 (29.5%) and 37/76 (48.7%); OR, 2.262; 95% CI, 1.028–4.977; *p* = 0.040). A lack of cost-free drug support (progesterone) for women with short cervices was identified as a barrier to preventing preterm births (30/44 (68.2%) and 32/76 (42.1%); OR, 0.339; 95% CI, 0.155–0.741; *p* = 0.006). Conclusions: Many physicians are unconvinced that measuring cervical length prevents premature births, and are unsure about the correct measurement procedures. There is a lack of government funding for hormone-usage programs.

## 1. Introduction

Globally, the chief cause of death among children younger than 5 years of age is a result of complications associated with preterm birth. In 2016, such complications accounted for 35% of neonatal deaths and approximately 16% of deaths in children up to age 5 worldwide [1]. Preterm births represented 12.98% of deliveries at Siriraj Hospital, Thailand, in 2008, [2] and the estimated global preterm birth rate for 2014 was 10.6% [3].

Premature babies who survive are at high risk of many short- and long-term illnesses. Common complications include respiratory distress syndrome, bronchopulmonary dysplasia, necrotizing enterocolitis, sepsis, periventricular leukomalacia, seizures, intraventricular hemorrhage, cerebral palsy, infections, feeding difficulties, and hypoxic-ischemic encephalopathy, as well as visual and hearing problems [4,5].

The financial burden of premature deliveries is substantial for the healthcare system and for parents [6]. About two-thirds of premature deliveries result in financial and emotional difficulties for the parents [6]. Research suggests that only a small number (7–15%) of spontaneous preterm births occur in women who have had a previous preterm delivery [7,8]. However, pregnant women with a cervical length of less than 25 mm between 18 and 24 weeks of gestation have an increased risk of premature birth [9]. Consequently, there is a need to have effective cervical length screening strategies in order to prevent preterm births. One study found that the vaginal administration of progesterone before 34 weeks of gestation reduced the incidence of preterm births in approximately 45% of women with a short cervix [10].

There are an estimated 15,000 cases of preterm birth annually in Thailand [11]. The expenditure on preterm neonatal hospital care has been calculated to be in the order of THB 170,000/case (USD 5312/case), which equates to a total of THB 255,000,000/year (USD 79,680,000/year). These figures exclude the long-term care costs incurred following hospital discharge [11].

The Society for Maternal-Fetal Medicine [12], the American College of Obstetricians and Gynecologists [13], the National Institute for Health and Care Excellence, [14] and the Thai Ministry of Public Health [15] support cervical length screening. They recommend that screening be performed between 20 and 24 weeks of gestation in order to identify women at risk of preterm delivery. The International Federation of Gynaecology and Obstetrics has recommended that screening be performed for all pregnant women [16]. As part of Thailand’s national health protocols for preventing preterm births, cervical length measurements should be taken from all women between 20 and 24 weeks of gestation [16]. Women with a short cervix (<25 mm) are deemed to be at high risk for preterm delivery. According to the guidelines of the Royal College of Obstetricians of Thailand, micronized progesterone vaginal suppositories are indicated for preventing preterm delivery [16].

Obstetricians play a significant role in the antenatal care of pregnant women. Pregnant mothers generally trust their obstetricians to inform them of all the necessary details relating to screening for and preventing preterm births. Unfortunately, despite Thailand’s universal screening policy being in force for over 4 years, its implementation has been largely unsuccessful: most pregnant women do not undergo screening. This research aimed to determine the obstacles to the performance of screening for preventing preterm deliveries from physicians’ perspectives.

## 2. Materials and Methods

This was a prospective, descriptive, exploratory cross-sectional study on physicians’ opinions and perspectives. It drew upon structured interviews and questionnaires derived from deep interviewing. The study was conducted at tertiary hospitals throughout the 6 regions of Thailand (northern, northeastern, southern, eastern, western, and central) from September 2019 to August 2020. Before the commencement of the research, ethics approval was obtained from the Siriraj Ethics Committee of the Faculty of Medicine, Siriraj Hospital, and the work was registered at the Thai Clinical Trials Registry.

To ensure the dataset was adequately sized to reach sufficiency for the details of barriers, we used a proportion for the results of interest of 50% (*p* = 0.5), an estimation error of ≤5%, and a 95% confidence level (type I error = 0.05; 2-sided). After factoring in the proportion of 1 physician to 3 patients in the healthcare system, it was determined that 120 physicians had to be questioned.
$$n=\frac{{\left(1.96\right)}^{2}\left(0.51\right)\left(1-0.5\right)}{0.05^{2}}=\frac{360}{3}=120$$

The research was divided into 3 phases.

### 2.1. In-Depth Interviews

This phase collected information in the following 4 areas:(1)General physician information;(2)Physician attitudes to the performance of cervical length measurements and the provision of care for preterm births;(3)The decision-making process for performing measurements;(4)Frustrations experienced when deciding whether to prevent preterm labor when a short cervix is detected.

The interviewers traveled to tertiary hospitals in various provinces throughout all 6 regions of Thailand. In each region, 4 to 6 hospitals were randomly selected, and one physician working in each hospital was interviewed. If the doctor was unwilling to participate, another hospital in the same region was selected. A total of 20 physicians were interviewed.

Physicians willing to participate in the research project were invited to a private counseling room. After the details of the proposed project were described, the physicians were invited to ask questions and given time to consider whether they wished to proceed with their formal enrollment in the trial. The physicians were informed that they could decline to participate in the research and, if they agreed to proceed, could withdraw at any stage. Twenty physicians subsequently volunteered as research subjects.

It was requested that the participants sign an informed consent form before being interviewed. Permission was obtained from each participant for the structured interview to be audio recorded. The subjects initially completed an attitude assessment questionnaire: this dealt with the methods used to measure cervical length and the assessment of the degree of care to be provided in the event of preterm births. Several other aspects were then investigated in the interview—one related to frustrations that might be felt before performing a cervical length measurement. The total time from the commencement of the questionnaire until the completion of the comprehensive interview was approximately 30 min. The data integrity of the research questions was later verified.

### 2.2. Development and Validation of the Questionnaire

The data obtained from the questionnaire and in-depth interviews were analyzed in order to determine the means and standard deviations. This enabled the questionnaire and interview questions to be refined. The revised questionnaire and interview questions were tested for validity and reliability before their use in the next phase. The questionnaire’s validity was checked by a statistician specialized in questionnaire construction and identifying double-barreled, confusing, and leading questions. To assess the test–retest reliability of the questionnaire, the same respondents completed the questionnaire again 1 month after first completing it. The data obtained from the questionnaire are detailed in “Appendix A”.

### 2.3. Administration of the Questionnaire

During the last phase of the study, the validated questionnaires were distributed to 120 physicians in tertiary hospitals. We wanted to find the barriers to cervical length screening at the level of tertiary hospitals, where adequate numbers of ultrasound machines and obstetricians are available. The tendency for Thai primary and secondary hospitals to not have adequate numbers of obstetricians or ultrasound machines is a known barrier to screening in the country.

The questionnaires were sent to various hospitals throughout the 6 regions of Thailand. In all, 24 tertiary hospitals were randomly selected using block randomization. The questionnaires were sent via registered mail. Each hospital was contacted to ensure that the questionnaires were completed and returned.

### 2.4. Statistical Analysis

Demographic data are summarized using descriptive statistics. Categorical data are presented as numbers and percentages, and continuous data are reported as mean ± standard deviation, or median and range. The statistical analyses were performed with PASW Statistics for Windows (version 18; SPSS Inc, Chicago, IL, USA). Hierarchical cluster analysis was employed because the variables used to group cases were “Yes” and “No.” Group comparisons were made with independent *t*-tests, Mann–Whitney U tests, and Chi^2^ tests.

## 3. Results

The Phase I interviews of the 20 physicians in tertiary hospitals throughout Thailand revealed that 0% to 7% of pregnant women underwent cervical length screening. The preterm birth rate also ranged from 9% to 15% (Figure 1).

All questionnaires sent to the 120 participants were returned (a 100% return rate). Table 1 presents the personal information of two clusters of physicians: those perceiving that preterm births present a low to moderate level of problems, and those considering that preterm births present a high level of problems.

Of the 120 respondents, 108 physicians reported having performed cervical length screening, while 12 doctors had never conducted the screening. We found that screening was performed in conjunction with other work at most hospitals due to supportive policies (odds ratio [95% CI], 1.742 (0.105–28.840); *p* < 0.01; Table 2). Encouragement was given by local Maternal and Child Health Boards for implementing cervical length screening programs at the hospitals (odds ratio [95% CI], 1.742 (0.105–28.840); *p* < 0.01; Table 2).

Of the 120 respondents, 63 opined that preterm births have severe consequences, whereas 57 stated that the births have low to moderate consequences. We found that most hospitals had enough obstetricians who could accurately perform cervical length measurements (odds ratio [95% CI], 4.261 (1.312–13.834); *p* < 0.011; Table 3).

Of the 120 respondents, 108 people indicated that their hospital had an action plan for preventing preterm births, while 18 doctors stated that their hospitals did not have such a plan. The factors significantly associated with an action plan are presented in Table 4.

Even though local Maternal and Child Health Boards supported the implementation of programs for cervical length screening for preterm birth prevention, the surveyed doctors did not think that cervical length screening plays a role in preventing preterm births (odds ratio [95% CI], 4.615 (1.482–14.373); *p* < 0.005; Table 5). The doctors were significantly unsure about the correct procedures for the measurements (odds ratio [95% CI], 2.262 (1.028–4.977); *p* < 0.040; Table 5).

The significant problems when screening is performed for high-risk pregnant women are the skills and knowledge of the physicians and the knowledge of the patients (Table 6). Providing knowledge and skills relating to cervical length measurements for doctors who perform routine work is essential so that they can become certificated and undertake examinations confidently (odds ratio [95% CI], 2.400 (1.130–5.098); *p* = 0.022; Table 6).

Hierarchical cluster analysis was performed by grouping cases of a similar nature. We assumed that the doctors who reported that a heavy workload was a major barrier were the same as those who mentioned a lack of government funding. The results of the cluster analysis were placed into two groups. A comparison of the respondents’ answers is given in Table 7. The answers with statistical significance were “other urgent and necessary tasks”, “excessive routine tasks”, and “insufficient number of personnel to support the performance of the procedure”.

## 4. Discussion

Our research found that the rate of cervical length screening at 20 tertiary centers was very low. About 90% of obstetricians were allowed to perform screening even if they had not received formal certification in the procedure. However, they required formal training to develop the knowledge and skills for cervical length measurements. Doing so would enable them to become certified and undertake examinations confidently while performing their routine work.

The current effective preventative measure for preterm deliveries is the use of progesterone [17]. Much research has supported that obtaining cervical length measurements is an effective screening method for pregnant women with short cervices. The procedure has also proven highly cost-effective with few risks [18,19]. Only a small proportion of women with preterm births have risk factors, and many preterm deliveries occur in nulliparous women. Therefore, universal transvaginal cervical length screening has been recommended in order to identify women prone to preterm birth [20].

One of the core barriers to the full implementation of universal screening in Thailand is the excessive volume of routine, urgent, and necessary tasks performed by physicians and nurses. Other perceived major barriers are the following:(1)Some physicians do not believe that the provision of universal screening justifies the requisite labor and funding.(2)There is inadequate funding by government agencies for both screening and the provision of cost-free progesterone.

Therefore, careful reconsideration of the need to perform universal screening is warranted.

Cervical length measurements can be safely performed during fetal structural assessments at 20 to 24 weeks of gestation. A transabdominal cervical length measurement should be offered to pregnant women with strong reservations about undergoing a transvaginal measurement [21,22]. Unfortunately, transabdominal measurements can be used only for some pregnant women [23]. When the procedure is performed, the cervical length will be longer than that determined by a transvaginal measurement. This is because the pregnant woman must have a full bladder in order to enable the ultrasound operator to obtain a clear field of view [23].

Cervical measurements are currently the most effective method, and transabdominal measurements should be reserved for women reluctant to undergo a transvaginal assessment. Regarding the cost-effectiveness of screening programs, transabdominal ultrasound should be performed for low-risk women during a fetal anatomy survey at 19 or 20 weeks of gestation, while the more accurate but relatively costly transvaginal ultrasound may be worthwhile for high-risk populations [24]. This approach has two benefits: First, the additional costs associated with transvaginal screening can be avoided [25]. Second, using dual methodologies improves the possibility that screening can be affordably performed for all pregnant women.

The vaginal administration of progesterone to women with a cervical length of ≤25 mm significantly reduces the risk of preterm birth [26]. The free supply of progesterone should be considered a national policy to prevent preterm births. However, one of many barriers to universal screening is the limited knowledge of the physicians involved in counseling pregnant women. If physicians do not believe in prevention strategies, the need for universal screening, or the benefits of progesterone treatment, screening utilization will be impaired [27,28]. The Maternal and Child Health Board can facilitate the implementation of universal cervical length screening. On the one hand, it could support the funding for training medical personnel in measuring cervical length, as well as the organization of regular training courses on preterm birth prevention for physicians and patients. Furthermore, it could also be responsible for providing the related medicines and medical supplies to all hospitals. These actions would ensure that screening is fully implemented, thereby reducing the preterm birth rate.

A physician’s expertise in taking measurements markedly affects the results of cervical examinations. Incorrect results may lead to unnecessary treatment or missed opportunities in preventing preterm births by administering vaginal progesterone. The performance quality and the learning curve associated with obtaining accurate measurements are critically important [29,30]. Providing cervical measurement training to physicians will likely increase the screening rate in many centers.

Our study aimed to identify barriers to cervical length screening at tertiary centers in Thailand, where adequate human, material, and drug resources are available. A limitation is that there was a relatively small number of participants (120 doctors), all of whom worked at tertiary-level hospitals. Therefore, their questionnaire responses may only partially reflect the views of physicians at the many primary-, secondary-, and tertiary-care hospitals throughout Thailand. However, we ameliorated this limitation by randomly assigning the questionnaire to hospitals throughout all six regions of Thailand. The recommendations of our study can be modified for implementation at primary and secondary centers.

## 5. Conclusions

There are two major obstacles to achieving universal cervical length measurements. One is the skepticism of physicians that such screenings can stave off preterm births. The other is government agencies’ lack of monetary support for hormone usage. Physicians are also unsure about the correct procedures for obtaining cervical length measurements. In order to overcome these barriers:Workloads should be reduced by extending the screening program to secondary centers.Government funding should be provided for progestogen usage.Physicians should be trained in transabdominal and transvaginal ultrasound.

## Figures and Tables

**Figure 1 healthcare-11-01039-f001:**
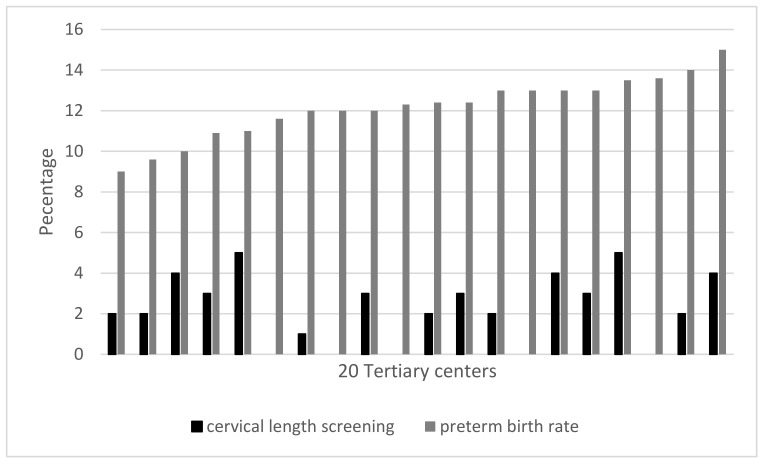
Proportions of women undergoing cervical length screening and preterm birth rates at 20 tertiary centers revealed via the Phase I in-depth interviews.

**Table 1 healthcare-11-01039-t001:** Personal information (more than one option was able to be selected).

Details of Personal Information	Total (n = 120)	Cluster #1 (n = 57)	Cluster #2 (n = 63)	*p*
Age (years) (mean ± SD [range])	38.8 ± 7.9 (26, 60)	39.5 ± 8.7 (26, 60)	38.2 ± 7.1 (29, 57)	0.366
Years since graduating with a medical degree (mean ± SD [range])	14.3 ± 7.8 (2, 36)	15.1 ± 8.6 (2, 36)	13.6 ± 7.1 (5, 31)	0.309
Postgraduate Diploma in Obstetrics and Gynecology n (%)	118 (98.3%)	55 (96.5%)	63 (100.0%)	0.224
Postgraduate Diploma in Maternal and Fetal Medicine (n [%])	25 (20.8%)	13 (22.8%)	12 (19.0%)	0.613
Years worked in the field of obstetrics and gynecology (mean ± SD [range])	9.1 ± 7.6 (1, 30)	10.1 ± 8.7 (1, 30)	8.1 ± 6.4 (1, 25)	0.159
Years worked in the position of head of department/unit of obstetrics and gynecology (median [range]	3.0 (0.1, 30.0)	2.0 (0.5, 30.0)	4.0 (0.1, 24.0)	0.196
Total bed capacity of the hospital (n [%]):				
120–300	7 (5.8%)	3 (5.3%)	4 (6.3%)	
300–500	52 (43.3%)	23 (40.4%)	29 (46.0%)	0.758
>500	61 (50.9%)	31 (54.4%)	30 (47.6%)	
Duties other than administrative work (n [%]):				
Teaching	29 (24.2%)	13 (22.8%)	16 (25.4%)	0.741
Service	94 (78.3%)	43 (75.4%)	51 (81.0%)	0.464
Research	21 (17.5%)	11 (19.3%)	10 (15.9%)	0.622
Other (e.g., Adolescent Clinic or Maternal and Child Health Board)	11 (9.2%)	7 (12.3%)	4 (6.3%)	0.261

Cluster #1: preterm births present a low to moderate level of problems. Cluster #2: preterm births present a high level of problems.

**Table 2 healthcare-11-01039-t002:** Factors associated with implementing the Ministry of Public Health policy for universal cervical length screening in order to prevent preterm births.

Factors	Screening Tests Are Performed (n = 102)	No Screening Tests Are Performed (n = 18)	Odds Ratio (95% CI)	*p*
Problems related to healthcare managers				
Concrete hospital policy for cervical length screening to prevent preterm births:				<0.001
No policy	17 (16.7%)	16 (88.9%)	50.824 (6.270, 411.942)	
Screening is done in parallel with other duties (eg, teaching or research)	31 (30.4%)	1 (5.6%)	1.742 (0.105, 28.840)	
There is a specific operating policy	54 (52.9%)	1 (5.6%)	1	
Working group or committee established to implement preterm birth prevention:				0.295
No assignment	30 (29.4%)	7 (38.9%)	3.383 (0.648, 17.657)	
Only some personnel are assigned	43 (42.2%)	9 (50.0%)	3.035 (0.611, 15.076)	
Working group established	29 (28.4%)	2 (11.1%)	1	
Action plan to prevent preterm births in the hospital:				0.387
No	9 (8.8%)	3 (16.7%)	1	
Yes	93 (91.2%)	15 (83.3%)	0.484 (0.117, 1.994)	
Have a role as a working physician in formulating policies relating to preterm birth prevention:				0.567
No	17 (16.7%)	4 (22.2%)	1	
Yes	85 (83.3%)	14 (77.8%)	0.700 (0.205, 2.388)	
Encouragement is given by the Maternal and Child Health Board for the conduct of a cervical length screening program at the hospital:				0.002
No	30 (29.4%)	12 (66.7%)	4.800 (1.649, 13.973)	
Yes	72 (70.6%)	6 (33.3%)	1	
Support is provided by the Maternal and Child Health Board for the implementation of a program of preterm birth prevention:				
Micronized progesterone vaginal soft-gel capsules (utrogestan)	43 (42.2%)	3 (16.7%)	0.274 (0.075, 1.007)	0.074
Progesterone pessaries (cyclogest)	-	-	-	-
17-OHPC (proluton depot)	34 (33.3%)	4 (22.2%)	0.571 (0.175, 1.869)	0.35
Funding for training of medical personnel in cervical length measurement	20 (19.6%)	0 (0.0%)	-	0.086
Funding for the purchase of ultrasound equipment	13 (12.7%)	1 (5.6%)	0.403 (0.049, 3.286)	0.691
Perception of the current role of the Maternal and Child Health Board in a cervical length screening program to prevent preterm births:				0.015
No role at all	17 (16.7%)	7 (38.9%)	14.824 (1.687, 130.248)	
Limited role	49 (48.0%)	10 (55.6%)	7.347 (0.900, 60.006)	
Very active	36 (35.3%)	1 (5.6%)	1	
Hospital regularly employs an adequate number of obstetricians to meet workloads:				0.75
No	38 (37.3%)	6 (33.3%)	1	
Yes	64 (62.7%)	12 (66.7%)	1.188 (0.412, 3.424)	
Hospital has obstetricians who can accurately perform cervical length measurements:				0.011
No and Yes, but not enough	46 (45.1%)	14 (77.8%)	4.261 (1.312, 13.834)	
Yes, enough	56 (54.9%)	4 (22.2%)	1	
Hospital has a person responsible for providing information on a preterm birth prevention program (Project Manager):				0.239
No	47 (46.1%)	11 (61.1%)	1.839 (0.660, 5.123)	
Yes	55 (53.9%)	7 (38.9%)	1	
Hospital has a specific budget for cervical length measurement screening:				0.213
No	89 (87.3%)	18 (100.0%)		
Yes (funds are sourced from the district budget)	13 (12.7%)	0 (0.0%)	-	
Hospital has enough ultrasound machines that can be used for routine tasks:				0.832
Not enough	37 (36.3%)	7 (38.9%)	1	
Enough	65 (63.7%)	11 (61.1%)	0.895 (0.319, 2.506)	
Hospital has an ultrasound machine that can be used specifically for a cervical length measurement screening program:				0.939
No	50 (49.0%)	9 (50.0%)	1	
Yes	52 (51.0%)	9 (50.0%)	0.962 (0.353, 2.619)	
Problems related to physicians				
Insufficient number of doctors available to perform the procedure	42 (41.2%)	9 (50.0%)	1.429 (0.523, 3.901)	0.485
Doctors have other urgent and necessary tasks	54 (52.9%)	9 (50.0%)	0.889 (0.326, 2.422)	0.818
Doctors have excessive routine tasks	67 (65.7%)	11 (61.1%)	0.821 (0.292, 2.304)	0.708
Doctors do not think that premature births are such a severe problem that the scheme is required	4 (3.9%)	0 (0.0%)	-	1
Doctors do not think that cervical length screening plays a role in preventing preterm births		4 (22.2%)	0.929 (0.279, 3.088)	1
Doctors do not believe that universal cervical length screening to prevent preterm births can justify the requisite labor and funding	31 (30.4%)	6 (33.3%)	1.145 (0.394, 3.328)	0.803
Doctors are unsure about the correct procedures for the measurements	42 (41.2%)	8 (44.4%)	1.143 (0.416, 3.137)	0.795
Problems related to other personnel, such as nurses and administrative staff				
Insufficient number of personnel to support the performance of the procedure	63 (61.8%)	12 (66.7%)	1.238 (0.430, 3.567)	0.692
There are other tasks that are more urgent	33 (32.4%)	5 (27.8%)	0.804 (0.265, 2.444)	0.7
The staff already have an excessive volume of routine tasks to perform	64 (62.7%)	8 (44.4%)	0.475 (0.173, 1.308)	0.144
Lack of confidence that the collecting, recording, and analyzing of the data by non-medical personnel will be accurate	42 (41.2%)	9 (50.0%)	1.429 (0.523, 3.901)	0.485
Problems related to the hospital				
Hospital administrators ignore the issue	16 (15.7%)	5 (27.8%)	2.067 (0.647, 6.603)	0.309
Lack of support for operating funds from government agencies	57 (55.9%)	10 (55.6%)	0.987 (0.360, 2.705)	0.979
Lack of cost-free drug support (progesterone) for pregnant women with short cervices to prevent preterm births	50 (49.0%)	12 (66.7%)	2.080 (0.725, 5.968)	0.167

**Table 3 healthcare-11-01039-t003:** Factors associated with awareness of the severity of the consequences of preterm births.

Factors	Level of the Awareness of Severity of the Consequences		Odds Ratio (95% CI)	*p*
	Low to Moderate (n = 57)	High (n = 63)		
Problems related to healthcare managers				
Concrete hospital policy for cervical length screening to prevent preterm births:				0.205
- No policy	20 (35.1%)	13 (20.6%)	1.00	
- Screening is done in parallel with other duties (e.g., teaching or research)	14 (24.6%)	18 (28.6%)	1.978 (0.737, 5.311)	
- There is a specific operating policy	23 (40.4%)	32 (50.8%)	2.140 (0.888, 5.161)	
Working group or committee established to implement preterm birth prevention:				0.629
- No assignment	20 (35.1%)	17 (27.0%)	1.00	
- Only some personnel are assigned	23 (40.4%)	29 (46.0%)	1.483 (0.636, 3.460)	
- Working group established	14 (24.6%)	17 (27.0%)	1.429 (0.548, 3.725)	
Action plan to prevent preterm births in the hospital:				0.670
- No	5 (8.8%)	7 (11.1%)	1.00	
- Yes	52 (91.2%)	56 (88.9%)	0.769 (0.230, 2.575)	
Have a role as a working physician in formulating policies relating to preterm birth prevention:				0.342
- No	8 (14.0%)	13 (20.6%)	1.00	
- Yes	49 (86.0%)	50 (79.4%)	0.628 (0.239, 1.648)	
Encouragement is given by the Maternal and Child Health Board for the conduct of a cervical length screening program at the hospital:				0.716
- No	19 (33.3%)	23 (36.5%)	1.00	
- Yes	38 (66.7%)	40 (63.5%)	0.870 (0.410, 1.846)	
Support is provided by the Maternal and Child Health Board for the implementation of a program of preterm birth prevention:				
- Micronized progesterone vaginal soft-gel capsules (Utrogestan)	23 (40.4%)	23 (36.5%)	0.850 (0.407, 1.776)	0.665
- Progesterone pessaries (cyclogest)	-	-	-	-
- 17-OHPC (proluton depot)	22 (38.6%)	16 (25.4%)	0.542 (0.249, 1.180)	0.121
- Funding for training of medical personnel in cervical length measurement	5 (8.8%)	15 (23.8%)	3.250 (1.098, 9.623)	0.027
- Funding for the purchase of ultrasound equipment	5 (8.8%)	9 (14.3%)	1.733 (0.545, 5.516)	0.347
Perception of the current role of the Maternal and Child Health Board in a cervical length screening program to prevent preterm births:				0.175
- No role at all	14 (24.6%)	10 (15.9%)	1.00	
- Limited role	23 (40.4%)	36 (57.1%)	2.191 (0.834, 5.755)	
- Very active	20 (35.1%)	17 (27.0%)	1.190 (0.422, 3.359)	
Hospital regularly employs an adequate number of obstetricians to meet workloads:				0.025
- No	15 (26.3%)	29 (46.0%)	1.00	
- Yes	42 (73.7%)	34 (54.0%)	0.419 (0.194, 0.904)	
Hospital has obstetricians who can accurately perform cervical length measurements:				0.855
- No and Yes, but not enough	28 (49.1%)	32 (50.8%)	1.00	
- Yes, enough	29 (50.9%)	31 (49.2%)	0.935 (0.457, 1.915)	
Hospital has a person responsible for providing information on a preterm birth prevention program (Project Manager):				0.571
- No	26 (45.6%)	32 (50.8%)	1.00	
- Yes	31 (54.4%)	31 (49.2%)	0.813 (0.396, 1.666)	
Hospital has a specific budget for cervical length measurement screening:				0.490
- No	52 (91.2%)	55 (87.3%)	1.00	
- Yes (funds are sourced from the district budget)	5 (8.8%)	8 (12.7%)	1.513 (0.465, 4.923)	
Hospital has enough ultrasound machines that can be used for routine tasks:				0.240
- Not enough	24 (42.1%)	20 (31.7%)	1.00	
- Enough	33 (57.9%)	43 (68.3%)	1.564 (0.741, 3.300)	
Hospital has an ultrasound machine that can be used specifically for a cervical length measurement screening program:				0.721
- No	29 (50.9%)	30 (47.6%)	1.00	
- Yes	28 (49.1%)	33 (52.4%)	1.139 (0.556, 2.334)	
Problems related to physicians				
- Insufficient number of doctors available to perform the procedure	25 (43.9%)	26 (41.3%)	0.899 (0.436, 1.857)	0.774
- Doctors have other urgent and necessary tasks	32 (56.1%)	31 (49.2%)	0.757 (0.369, 1.554)	0.448
- Doctors have excessive routine tasks	34 (59.6%)	44 (69.8%)	1.567 (0.737, 3.332)	0.242
- Doctors do not think that premature births are such a severe problem that the scheme is required	3 (5.3%)	1 (1.6%)	0.290 (0.029, 2.874)	0.345
- Doctors do not think that cervical length screening plays a role in preventing preterm births	12 (21.1%)	16 (25.4%)	1.277 (0.544, 2.995)	0.574
- Doctors do not believe that universal cervical length screening to prevent preterm births can justify the requisite labor and funding	18 (31.6%)	19 (30.2%)	0.936 (0.431, 2.032)	0.866
- Doctors are unsure about the correct procedures for the measurements	21 (36.8%)	29 (46.0%)	1.462 (0.704, 3.039)	0.308
Problems related to other personnel, such as nurses and administrative staff				
- Insufficient number of personnel to support the performance of the procedure	35 (61.4%)	40 (63.5%)	1.093 (0.522, 2.291)	0.813
- There are other tasks that are more urgent	19 (33.3%)	19 (30.2%)	0.864 (0.400, 1.865)	0.709
- The staff already have an excessive volume of routine tasks to perform	29 (50.9%)	43 (68.3%)	2.076 (0.988, 4.361)	0.052
- Lack of confidence that the collecting, recording, and analyzing of the data by non-medical personnel will be accurate	24 (42.1%)	27 (42.9%)	1.031 (0.500, 2.129)	0.934
Problems related to the hospital				
- Hospital administrators ignore the issue	11 (19.3%)	10 (15.9%)	0.789 (0.307, 2.026)	0.622
- Lack of support for operating funds from government agencies	30 (52.6%)	37 (58.7%)	1.281 (0.622, 2.638)	0.502
- Lack of cost-free drug support (progesterone) for pregnant women with short cervices to prevent preterm births	30 (52.6%)	32 (50.8%)	0.929 (0.454, 1.903)	0.841

**Table 4 healthcare-11-01039-t004:** Factors associated with an action plan for preventing preterm births in the hospital.

Factors	Action Plan to Prevent Preterm Births in the Hospital		Odds Ratio (95% CI)	*p*
	No (n = 12)	Yes (n = 108)		
Problems related to physicians				
- Insufficient number of doctors available to perform the procedure	6 (50.0%)	45 (41.7%)	0.714 (0.216, 2.359)	0.580
- Doctors have other urgent and necessary tasks	9 (75.0%)	54 (50.0%)	0.333 (0.086, 1.299)	0.100
- Doctors have excessive routine tasks	9 (75.0%)	69 (63.9%)	0.590 (0.151, 2.308)	0.444
- Doctors do not think that premature births are such a severe problem that the scheme is required	0 (0.0%)	4 (3.7%)	-	1.000
- Doctors do not think that cervical length screening plays a role in preventing preterm births	5 (41.7%)	23 (21.3%)	0.379 (0.110, 1.305)	0.148
- Doctors do not believe that universal cervical length screening to prevent preterm births can justify the requisite labor and funding	5 (41.7%)	32 (29.6%)	0.589 (0.174, 1.996)	0.511
- Doctors are unsure about the correct procedures for the measurements	3 (25.0%)	47 (43.5%)	2.311 (0.593, 9.014)	0.217
Problems related to other personnel, such as nurses and administrative staff				
- Insufficient number of personnel to support the performance of the procedure	8 (66.7%)	67 (62.0%)	0.817 (0.231, 2.885)	1.000
- There are other tasks that are more urgent	4 (33.3%)	34 (31.5%)	0.919 (0.259, 3.263)	1.000
- The staff already have an excessive volume of routine tasks to perform	9 (75.0%)	63 (58.3%)	0.467 (0.120, 1.821)	0.358
- Lack of confidence that the collecting, recording, and analyzing of the data by non-medical personnel will be accurate	5 (41.7%)	46 (42.6%)	1.039 (0.310, 3.481)	0.951
Problems related to the hospital				
- Hospital administrators ignore the issue	3 (25.0%)	18 (16.7%)	0.600 (0.148, 2.436)	0.439
- Lack of support for operating funds from government agencies	5 (41.7%)	62 (57.4%)	1.887 (0.563, 6.324)	0.298
- Lack of cost-free drug support (progesterone) for pregnant women with short cervices to prevent preterm births	7 (58.3%)	55 (50.9%)	0.741 (0.221, 2.481)	0.626

**Table 5 healthcare-11-01039-t005:** Factors associated with support provided by the Maternal and Child Health Board for implementation of preterm birth prevention programs.

Factors	Support Provided by the Maternal and Child Health Board for the Implementation of a Program of Preterm Birth Prevention		Odds Ratio (95% CI)	*p*
	No (n = 44)	Yes (n = 76)		
Problems related to physicians				
- Insufficient number of doctors available to perform the procedure	22 (50.0%)	29 (38.2%)	0.617 (0.291, 1.307)	0.206
- Doctors have other urgent and necessary tasks	23 (52.3%)	40 (52.6%)	1.014 (0.482, 2.134)	0.970
- Doctors have excessive routine tasks	30 (68.2%)	48 (63.2%)	0.800 (0.364, 1.758)	0.578
- Doctors do not think that premature births are such a severe problem that the scheme is required	0 (0.0%)	4 (5.3%)	-	0.295
- Doctors do not think that cervical length screening plays a role in preventing preterm births	4 (9.1%)	24 (31.6%)	4.615 (1.482, 14.373)	0.005
- Doctors do not believe that universal cervical length screening to prevent preterm births can justify the requisite labor and funding	9 (20.5%)	28 (36.8%)	2.269 (0.952, 5.405)	0.061
- Doctors are unsure about the correct procedures for the measurements	13 (29.5%)	37 (48.7%)	2.262 (1.028, 4.977)	0.040
Problems related to other personnel, such as nurses and administrative staff				
- Insufficient number of personnel to support the performance of the procedure	28 (63.6%)	47 (61.8%)	0.926 (0.429, 1.998)	0.845
- There are other tasks that are more urgent	14 (31.8%)	24 (31.6%)	0.989 (0.445, 2.196)	0.978
- The staff already have an excessive volume of routine tasks to perform	24 (54.5%)	48 (63.2%)	1.429 (0.672, 3.038)	0.353
- Lack of confidence that the collecting, recording, and analyzing of the data by non-medical personnel will be accurate	17 (38.6%)	34 (44.7%)	1.286 (0.603, 2.740)	0.515
Problems related to the hospital				
- Hospital administrators ignore the issue	8 (18.2%)	13 (17.1%)	0.929 (0.352, 2.453)	0.881
- Lack of support for operating funds from government agencies	28 (63.6%)	39 (51.3%)	0.602 (0.281, 1.290)	0.190
- Lack of cost-free drug support (progesterone) for pregnant women with short cervices to prevent preterm births	30 (68.2%)	32 (42.1%)	0.339 (0.155, 0.741)	0.006

**Table 6 healthcare-11-01039-t006:** Relevant problems when screening is performed for high-risk pregnant women.

Relevant Problems	There Are Problems When Screening Is Performed for the Target Group (Pregnant Women Who Are at Risk):		Odds Ratio (95% CI)	*p*
	No (n = 46)	Yes (n = 74)		
Relevant to physicians or related individuals				
- Add/request additional doctors who have the potential to screen cervical lengths using various methods	12 (26.1%)	30 (40.5%)	1.932 (0.863, 4.322)	0.107
- Provide regular training to physicians to enable them to confidently measure cervical lengths	22 (47.8%)	46 (62.2%)	1.792 (0.851, 3.776)	0.123
- Provide knowledge and skills relating to cervical length measurements for doctors who perform routine work so that they can become certificated and undertake examinations confidently	20 (43.5%)	48 (64.9%)	2.400 (1.130, 5.098)	0.022
- Provide reliable research results/demonstrations of the procedure/examples of screening results, and present doctors/nurses/other stakeholders with a detailed and convincing case for the cost-effectiveness of implementing universal cervical length screening	19 (41.3%)	29 (39.2%)	0.916 (0.433, 1.938)	0.818
- Reduce extraneous duties for doctors	18 (39.1%)	33 (44.6%)	1.252 (0.952, 2.647)	0.556
Relevant to hospitals				
- Provide hospitals with adequate and regular funding from relevant agencies	35 (76.1%)	57 (77.0%)	1.054 (0.443, 2.509)	0.906
- Extend screening to community hospitals to relieve workloads at tertiary center	30 (65.2%)	54 (73.0%)	1.440 (0.651, 3.187)	0.367
- Educate patients about the benefits of cervical length measurements to prevent preterm births	24 (52.2%)	54 (73.0%)	2.475 (1.142, 5.363)	0.020

**Table 7 healthcare-11-01039-t007:** Cluster analysis by grouping physicians who stated that both heavy workloads and a lack of government funding were major barriers.

	Cluster #1 (n = 57)	Cluster #2 (n = 63)	Odds Ratio (95% CI)	*p*
Problems related to physicians				
- Insufficient number of doctors available to perform the procedure	19 (33.3%)	32 (50.8%)	2.065 (0.985, 4.326)	0.053
- Doctors have other urgent and necessary tasks	10 (17.5%)	53 (84.1%)	24.910 (9.533, 65.088)	<0.001
- Doctors have excessive routine tasks	25 (43.9%)	53 (84.1%)	6.784 (2.886, 15.945)	<0.001
- Doctors do not think that premature births are such a severe problem that the scheme is required	1 (1.8%)	3 (4.8%)	2.800 (0.283, 27.713)	0.621
- Doctors do not think that cervical length screening plays a role in preventing preterm births	12 (21.1%)	16 (25.4%)	1.277 (0.544, 2.995)	0.574
- Doctors do not believe that universal cervical length screening to prevent preterm births can justify the requisite labor and funding	18 (31.6%)	19 (30.2%)	0.936 (0.431, 2.032)	0.866
- Doctors are unsure about the correct procedures for the measurements	18 (31.6%)	32 (50.8%)	2.237 (1.061, 4.714)	0.033
Problems related to other personnel, such as nurses and administrative staff				
- Insufficient number of personnel to support the performance of the procedure	20 (35.1%)	55 (87.3%)	12.719 (5.070, 31.907)	<0.001
- There are other tasks that are more urgent	2 (3.5%)	36 (57.1%)	36.667 (8.210, 163.757)	<0.001
- The staff already have an excessive volume of routine tasks to perform	14 (24.6%)	58 (92.1%)	35.629 (11.923, 106.463)	<0.001
- Lack of confidence that the collecting, recording, and analyzing of the data by non-medical personnel will be accurate	25 (43.9%)	26 (41.3%)	0.899 (0.436, 1.857)	0.774
Problems related to the hospital				
- Hospital administrators ignore the issue	5 (8.8%)	16 (25.4%)	3.540 (1.204, 10.414)	0.017
- Lack of support for operating funds from government agencies	33 (57.9%)	34 (54.0%)	0.853 (0.414, 1.756)	0.665
- Lack of cost-free drug support (progesterone) for pregnant women with short cervices to prevent preterm births	31 (54.4%)	31 (49.2%)	0.813 (0.396, 1.666)	0.571

Cluster #1: physicians who stated that a heavy workload was a major barrier. Cluster #2: physicians who stated that a lack of government funding was a major barrier.

## Data Availability

The authors confirm that the data supporting the findings of this study are available within the article and its Appendix A.

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
