# Peer review of "Identifying the Barriers to Universal Cervical Length Screening for Preterm Birth Prevention at a Tertiary Hospital in Thailand (Physician Perspectives): Implementation Research"

_healthcare, 2023, doi:10.3390/healthcare11071039_

Round 1

Reviewer 1 Report

This prospective cohort study assessed the Thai “tertiary based” physicians’ perspectives on the barriers of cervical length screening to prevent preterm births. Overall, the topic is very relevant, and deserves to be published. But the manuscript is poorly structured, difficult to read, and the methodology followed is not always clear to the reader. In its current form this manuscript cannot be published. 

Specific comments: 

1.     General:

1.1.  There are multiple language use and syntax problems, and the manuscript would greatly benefit from formal copyediting. 

1.2.  Abstract should be limited to 200 words

2.     Introduction:

2.1.  Difficult to read due to poor language use, and cumbersome to follow that the authors are trying to say. 

2.2.  Very long winded, please keep to a coherent message on why the research was done. Thea authors states multiple times that universal cervical length screening should be done including being endorsed even by their local society.

3.     Methodology:

3.1.  The author states that this is an implementation study. But this is purely an exploratory cross-sectional study on physicians’ opinions/perspectives using structured interviews/questionnaires. Please correct. 

3.2.  Please explain on how the power calculation was done and why this was necessary. 

3.3.  Phase II: How was the questionaries validity and reliability tested? Was this pure face and content validity by an expert panel? 

3.4.  Phase III: Why was only tertiary hospitals included if the authors wanted to assess universal screening? Why were the questionaries not sent to all obstetricians? 

4.     Results:

4.1.  In general, there is a lot of duplication of data between tables and text. Please refrain from doing this. Results should either be given in summarized tables or only in short text from.

4.2.   Did the authors have a 100% response rate (120/120). This is unheard of for questionnaire responses. Please elaborate. 

4.3.  Line 158: Start with Table 38.8 years. Please correct into coherent sentence. 

4.4.  Table 1 lists two clusters. This clusters/group allocation should be explained in the methods and included in a footnote. 

4.5.  The reader is flooded with too many observations and leads to the impression of discrepant findings. Authors should consider only keeping important tables, that focusses on the implementation of universal cervical length screening, in the manuscript and move all the rest to supplementary materials. 

5.     Discussion:

5.1.  Line 290-295: The study summary as given here is not supported by the results. Figure 1 indicates very low rate of cervical length screening, with only 46% of the respondents had a policy of cervical length screening, Furthermore, over 90% could perform cervical length screening and over 80% assist with policies yet 35% says that universal cervical length screening does not reduce preterm births. This is nothing to do with workload but hospital based protocols and education. 

5.2.  There is once again a lot of duplication between the discussion and introduction. Keep the discussion on the topic of the manuscript. I.e. what factors limit the implementation of this intervention. 

5.3.  Line 305-308: Do the authors now want to discredit universal screening? This is in direct contrast to why the authors have done the study. What are the authors trying to highlight?

5.4.  Paragraph 4 - 6: herein the authors now motivate for screening only in high risk pregnancies. Then goes on to motivate for transabdominal cervical assessments. Herein, the reader can not follow the arguments of the authors. 

5.5.  Line 355-361: This is probably the main constraint for implementation and the reader would have expected the authors to go more into detail about the funding, financing, and allocation of vaginal progesterone. Please expand on how the authors would approach these aspects. 

Author Response

Top of Form

Journal

Healthcare (ISSN 2227-9032)

Manuscript ID

healthcare-2196431

Type

Article

Title

Identifying the barriers to universal cervical length screening for preterm birth prevention at a tertiary hospital in Thailand (physician perspectives): implementation research

Authors

Saifon Chawanpaiboon * , Vitaya Titapant , Sanitra Anuwutnavin , Attapol Kanjanapongporn , Julaporn Pooliam

Section

Health Policy

Special Issue

Health and Social Care Policy

Abstract

Background: Cervical length screening is the method to predict preterm delivery. Objective: To identify physicians’ perspectives of the barriers to cervical length screening to prevent preterm births. Methods: A prospective, descriptive, implementation study was carried out at the official rooms, tertiary hospitals. Ethics approval was obtained from the Siriraj Ethics Committee of the Faculty of Medicine Siriraj Hospital, and the work was registered at the Thai Clinical Trials Registry (TCTR20190813003). Physicians at 52 tertiary hospitals throughout Thailand were recruited. In Phase I of this prospective descriptive implementation study, 20 physicians were interviewed. Phase II comprised questionnaire development and data validation. The questionnaire was administered to 120 Phase-III participants. Results: The 44 and 76 cases of physicians with and without support provided by the Maternal and Child Health Board were response. Doctors do not think that cervical length screening plays a role in preventing preterm births. [4/44(9.1%)/24/76 (31.6%)] ([OR] 4.615; 95% CI, (1.482, 14.373); P = 0.005). Doctors are unsure about the correct procedures for the measurements [13/44 (29.5%)/37/76 (48.7%)] ([OR] 2.262; 95% CI, (1.028, 4.977); P = 0.040). Lack of cost-free drug support (progesterone) for pregnant women with short cervices to prevent preterm births [30/44 (68.2%)/32/76 (42.1%)] ([OR] 0.339; 95% CI, (0.155, 0.741); P = 0.006). Cluster analysis by grouping of physicians who stated that both heavy workloads (57/120 cases) and lack of government funding (63/120 cases) were major barriers, both groups response that they have other urgent and necessary tasks [10/57 (17.5%)/53/63 (84.1%)] ([OR] 24.910; 95% CI, (9.533, 65.088); P < 0.001). Conclusions: From the physicians’ perspectives, the barriers to perform cervical length measurements are unbelieving that cervical length screening can prevent preterm. They are unsure about the correct procedures for the measurements and a lack of government funding for hormone-usage programs. High-risk pregnant women threatening preterm deliveries should be considered for screening.

Bottom of Form

Top of Form

Author's Reply to the Review Report (Reviewer 1)

Please provide a point-by-point response to the reviewer’s comments and either enter it in the box below or upload it as a Word/PDF file. Please write down "Please see the attachment." in the box if you only upload an attachment. An example can be found here.

* Author's Notes to Reviewer

FileEditViewInsertFormatToolsTableHelp

Paragraph

P

0 WORDS

Word / PDF

 or 

Bottom of Form

Top of Form

Review Report Form

Open Review

English language and style

( ) English very difficult to understand/incomprehensible
(x) Extensive editing of English language and style required
( ) Moderate English changes required
( ) English language and style are fine/minor spell check required
( ) I don't feel qualified to judge about the English language and style

Yes

Can be improved

Must be improved

Not applicable

Does the introduction provide sufficient background and include all relevant references?

( )

(x)

( )

( )

Are all the cited references relevant to the research?

( )

(x)

( )

( )

Is the research design appropriate?

( )

( )

(x)

( )

Are the methods adequately described?

( )

( )

(x)

( )

Are the results clearly presented?

( )

(x)

( )

( )

Are the conclusions supported by the results?

( )

( )

(x)

( )

Comments and Suggestions for Authors

This prospective cohort study assessed the Thai “tertiary based” physicians’ perspectives on the barriers of cervical length screening to prevent preterm births. Overall, the topic is very relevant, and deserves to be published. But the manuscript is poorly structured, difficult to read, and the methodology followed is not always clear to the reader. In its current form this manuscript cannot be published. 

Specific comments: 

  1. General:

1.1.  There are multiple language use and syntax problems, and the manuscript would greatly benefit from formal copyediting. 

Response:

Formal copyediting by native English speaker was done with the attachment of declared of certification of editing.

1.2.  Abstract should be limited to 200 words

Response:

Abstract was re-written to be 188 words.

  1. Introduction:

2.1.  Difficult to read due to poor language use, and cumbersome to follow that the authors are trying to say. 

Response:

Introduction was re-written.

2.2.  Very long winded, please keep to a coherent message on why the research was done. Thea authors states multiple times that universal cervical length screening should be done including being endorsed even by their local society.

Response:

Content in introduction part was re-written to be shorter.

  1. Methodology:

3.1.  The author states that this is an implementation study. But this is purely an exploratory cross-sectional study on physicians’ opinions/perspectives using structured interviews/questionnaires. Please correct. 

Response:

Methods was corrected to be “This was a prospective, descriptive, exploratory cross-sectional study on physicians’ opinions/perspectives using structured interviews/questionnaires which derived from deep interviewing”.

3.2.  Please explain on how the power calculation was done and why this was necessary. 

Response:

This survey study utilized questionnaires. To ensure an adequately sized dataset for sufficiency of details of barriers, with a proportion of the results of interest of 50% (P = 0.5), an estimation error of ≤ 5%, and a 95% confidence level (type I error = 0.05, 2-sided), the number of physicians needing to be surveyed was calculated to be 120 (the proportion of healthcare system by 1 physician to 3 patients).

  = 360/3 = 120

3.3.  Phase II: How was the questionaries validity and reliability tested? Was this pure face and content validity by an expert panel? 

Response:

Phase II: development and validation of questionnaire was rewritten as follow:

The data obtained from the questionnaire and in-depth interviews were analyzed to determine the means and standard deviations. This enabled the questionnaire and interview questions to be refined. The revised questionnaires and interview questions were tested for validity and reliability before being used in the next phase. The method for questionnaire reliability was test-retest reliability by giving the questionnaire to the same group of respondents at one month after revision. The questionnaire validity was checked by the statistician who was an expert on questionnaire construction and examined for double, confusing and leading questions. The data are detailed in “S1: Questionnaire for Physician’s Perspective”.

3.4.  Phase III: Why was only tertiary hospitals included if the authors wanted to assess universal screening? Why were the questionaries not sent to all obstetricians? 

Response:

We wanted to find the barriers of cervical length screening at the level of tertiary hospitals where adequate ultrasound machines and obstetricians were provided. While secondary or primary hospitals may not have adequate obstetricians and ultrasound machines which are the known main barriers in Thailand.

  1. Results:

4.1.  In general, there is a lot of duplication of data between tables and text. Please refrain from doing this. Results should either be given in summarized tables or only in short text from.

Response:

All are re-written.

4.2.   Did the authors have a 100% response rate (120/120). This is unheard of for questionnaire responses. Please elaborate. 

Response:

The 120 questionnaires were received.

4.3.  Line 158: Start with Table 38.8 years. Please correct into coherent sentence. 

Response:

All are re-written.

4.4.  Table 1 lists two clusters. This clusters/group allocation should be explained in the methods and included in a footnote. 

 Response:

Two clusters were analyzed in Table 1 and 7 which based on the results. We analyzed the physicians who aware for the level of severity of preterm births and the physicians who identified heavy workload as a key barrier the same as those who mentioned lack of government funding were the same group or not.

4.5.  The reader is flooded with too many observations and leads to the impression of discrepant findings. Authors should consider only keeping important tables, that focusses on the implementation of universal cervical length screening, in the manuscript and move all the rest to supplementary materials. 

Response:

Table 2-8 were moved to supplementary materials. 

  1. Discussion:

5.1.  Line 290-295: The study summary as given here is not supported by the results. Figure 1 indicates very low rate of cervical length screening, with only 46% of the respondents had a policy of cervical length screening, Furthermore, over 90% could perform cervical length screening and over 80% assist with policies yet 35% says that universal cervical length screening does not reduce preterm births. This is nothing to do with workload but hospital based protocols and education. 

Response: The summary was re-written as follow:

Our research found that the rate of cervical length screening among 20 tertiary centers was very low. About 90% of obstetricians can perform screening even if they have not received formal certification in the conduct of the procedure. They required the training for knowledge and skills relating to cervical length measurements so that they can become certificated and undertake examinations confidently, along with performing their routine work

5.2.  There is once again a lot of duplication between the discussion and introduction. Keep the discussion on the topic of the manuscript. I.e. what factors limit the implementation of this intervention. 

Response: The limited factors were added in the discussion.

5.3.  Line 305-308: Do the authors now want to discredit universal screening? This is in direct contrast to why the authors have done the study. What are the authors trying to highlight?

Response: This phrase was deleted. I tried to update available recent data. I agree with you that this phrase must be reconsidered to be re-written.

5.4.  Paragraph 4 - 6: herein the authors now motivate for screening only in high risk pregnancies. Then goes on to motivate for transabdominal cervical assessments. Herein, the reader can not follow the arguments of the authors. 

 Response: Paragraph 4 – 6 were deleted.

5.5.  Line 355-361: This is probably the main constraint for implementation and the reader would have expected the authors to go more into detail about the funding, financing, and allocation of vaginal progesterone. Please expand on how the authors would approach these aspects. 

Response: The Maternal and Child Health Board for the implementation of a program of preterm birth prevention by universal cervical length screening will be successful by supporting the funding for training of medical personnel in cervical length measurement and organizing regularly training course of preterm birth prevention for both physicians and patients. They should take a responsibility for providing medicines and medical supplies to all hospitals. Therefore, the implementation of universal cervical length screening would be reached resulted in reducing preterm birth rate.

Submission Date

17 January 2023

Date of this review

31 Jan 2023 16:30:04

Bottom of Form

Reviewer 2 Report

Dear authors;

This article has been absolutely dazzling. It has been a detailed analysis of a very important obsteric management. Thank you for your dedication and clarity in this study. The only thing that caught my attention is; It will reduce confusion if you redesign the tables and reorganize the places you see very important by combining the tables (3 or 4 tables maximum).

Author Response

Open Review

English language and style

( ) English very difficult to understand/incomprehensible
( ) Extensive editing of English language and style required
( ) Moderate English changes required
(x) English language and style are fine/minor spell check required
( ) I don't feel qualified to judge about the English language and style

Yes

Can be improved

Must be improved

Not applicable

Does the introduction provide sufficient background and include all relevant references?

(x)

( )

( )

( )

Are all the cited references relevant to the research?

(x)

( )

( )

( )

Is the research design appropriate?

(x)

( )

( )

( )

Are the methods adequately described?

(x)

( )

( )

( )

Are the results clearly presented?

(x)

( )

( )

( )

Are the conclusions supported by the results?

(x)

( )

( )

( )

Comments and Suggestions for Authors

Dear authors;

This article has been absolutely dazzling. It has been a detailed analysis of a very important obsteric management. Thank you for your dedication and clarity in this study. The only thing that caught my attention is; It will reduce confusion if you redesign the tables and reorganize the places you see very important by combining the tables (3 or 4 tables maximum).

Submission Date

17 January 2023

Date of this review

01 Feb 2023 11:51:32

Response:

Thank you for your revision. The total 13 tables were reduced to be 6 tables. The rest of 7 tables (previous Table 2-8) are in the supplementary materials.

Reviewer 3 Report

Most MFM subspecialists would agree cervical length measurement (CLM) can be a reliable way to predict preterm delivery where preterm labor is threatened. With this background, the current manuscript reports preliminary data from very confined sampling.

It is unlikely that the entire medical community in Thailand, which surely must be thousands of registered providers, can be studied by inputs from fewer than 200 participants.

Any study attempting to reflect attitudes of the entire national OB care environment, simply by auditing a few dozen individuals invites disbelief. It is disappointing that no study limitations are acknowledged whatsoever.

For CLM to be clinically useful, there is an expectation that preterm labor is first accurately diagnosed. In other words, random cervical measurements taken in an unselected OB population brings little value irrespective of other variables. It is implied that all queried personnel agree with this approach, although this is not specifically confirmed.

If the study was designed to audit cervical length policy at multiple facilities, why does the article title refer only to ‘a tertiary hospital’?

The paper is at best a pilot study. It would benefit from careful editing by someone fluent in medical English, as there are incomplete sentences and grammar errors throughout the work.

Author Response

Open Review

English language and style

( ) English very difficult to understand/incomprehensible
( ) Extensive editing of English language and style required
(x) Moderate English changes required
( ) English language and style are fine/minor spell check required
( ) I don't feel qualified to judge about the English language and style

Yes

Can be improved

Must be improved

Not applicable

Does the introduction provide sufficient background and include all relevant references?

( )

(x)

( )

( )

Are all the cited references relevant to the research?

(x)

( )

( )

( )

Is the research design appropriate?

( )

(x)

( )

( )

Are the methods adequately described?

( )

( )

(x)

( )

Are the results clearly presented?

( )

(x)

( )

( )

Are the conclusions supported by the results?

( )

( )

(x)

( )

Comments and Suggestions for Authors

Most MFM subspecialists would agree cervical length measurement (CLM) can be a reliable way to predict preterm delivery where preterm labor is threatened. With this background, the current manuscript reports preliminary data from very confined sampling.

It is unlikely that the entire medical community in Thailand, which surely must be thousands of registered providers, can be studied by inputs from fewer than 200 participants.

Any study attempting to reflect attitudes of the entire national OB care environment, simply by auditing a few dozen individuals invites disbelief. It is disappointing that no study limitations are acknowledged whatsoever.

For CLM to be clinically useful, there is an expectation that preterm labor is first accurately diagnosed. In other words, random cervical measurements taken in an unselected OB population brings little value irrespective of other variables. It is implied that all queried personnel agree with this approach, although this is not specifically confirmed.

If the study was designed to audit cervical length policy at multiple facilities, why does the article title refer only to ‘a tertiary hospital’?

The paper is at best a pilot study. It would benefit from careful editing by someone fluent in medical English, as there are incomplete sentences and grammar errors throughout the work.

Submission Date

17 January 2023

Date of this review

23 Feb 2023 17:01:01

Response:

Thank you for your revision. I have added the limitation of the study in the last paragraph as your suggestion.

The limitation of the study is the number of respondents of 120 people at tertiary hospitals which is not represented all physicians around Thailand. However this limitation was overcome by random sending the questionnaire throughout 6 regions of Thai-land. The objective of our study is to identify the barriers of cervical length screening in the tertiary centers where provided adequate human, materials and drug resource The results of our study can be modified implement to the primary and secondary centers. 

Formal copyediting by native English speaker was done with the attachment of declared of certification of editing.

Round 2

Reviewer 1 Report

Authors did go to great lengths to address all concerns and the manuscript now reads easily. 

Author Response

No additional correction.

Thank you for your revision.

Saifon Chawanpaiboon

Reviewer 3 Report

If the cover letter details are considered adequate, then the Editor may publish. 

Author Response

Open Review

English language and style

( ) English very difficult to understand/incomprehensible
( ) Extensive editing of English language and style required
(x) Moderate English changes required
( ) English language and style are fine/minor spell check required
( ) I don't feel qualified to judge about the English language and style

Yes

Can be improved

Must be improved

Not applicable

Does the introduction provide sufficient background and include all relevant references?

( )

(x)

( )

( )

Are all the cited references relevant to the research?

(x)

( )

( )

( )

Is the research design appropriate?

( )

(x)

( )

( )

Are the methods adequately described?

( )

( )

(x)

( )

Are the results clearly presented?

( )

(x)

( )

( )

Are the conclusions supported by the results?

( )

( )

(x)

( )

Comments and Suggestions for Authors

Most MFM subspecialists would agree cervical length measurement (CLM) can be a reliable way to predict preterm delivery where preterm labor is threatened. With this background, the current manuscript reports preliminary data from very confined sampling.

It is unlikely that the entire medical community in Thailand, which surely must be thousands of registered providers, can be studied by inputs from fewer than 200 participants.

Any study attempting to reflect attitudes of the entire national OB care environment, simply by auditing a few dozen individuals invites disbelief. It is disappointing that no study limitations are acknowledged whatsoever.

For CLM to be clinically useful, there is an expectation that preterm labor is first accurately diagnosed. In other words, random cervical measurements taken in an unselected OB population brings little value irrespective of other variables. It is implied that all queried personnel agree with this approach, although this is not specifically confirmed.

If the study was designed to audit cervical length policy at multiple facilities, why does the article title refer only to ‘a tertiary hospital’?

The paper is at best a pilot study. It would benefit from careful editing by someone fluent in medical English, as there are incomplete sentences and grammar errors throughout the work.

Submission Date

17 January 2023

Date of this review

23 Feb 2023 17:01:01

Response:

Thank you for your revision. I have added the limitation of the study in the last paragraph as your suggestion.

The limitation of the study is the number of respondents of 120 people at tertiary hospitals which is not represented all physicians around Thailand. However this limitation was overcome by random sending the questionnaire throughout 6 regions of Thai-land. The objective of our study is to identify the barriers of cervical length screening in the tertiary centers where provided adequate human, materials and drug resource The results of our study can be modified implement to the primary and secondary centers. 

Formal copyediting by native English speaker was done with the attachment of declared of certification of editing.

30/3/2023

Reply to editor

No additional correction.

Thank you very much for your revision.

Saifon Chawanpaiboon
